# Salivary Redox Homeostasis in Human Health and Disease

**DOI:** 10.3390/ijms231710076

**Published:** 2022-09-03

**Authors:** Beáta Čižmárová, Vladimíra Tomečková, Beáta Hubková, Anna Hurajtová, Jana Ohlasová, Anna Birková

**Affiliations:** Department of Medical and Clinical Biochemistry, Faculty of Medicine, Pavol Jozef Šafárik University in Košice, Trieda SNP 1, 040 11 Košice, Slovakia

**Keywords:** saliva, antioxidant, disease, homeostasis

## Abstract

Homeostasis is a self-regulatory dynamic process that maintains a stable internal environment in the human body. These regulations are essential for the optimal functioning of enzymes necessary for human health. Homeostasis elucidates disrupted mechanisms leading to the development of various pathological conditions caused by oxidative stress. In our work, we discuss redox homeostasis and salivary antioxidant activity during healthy periods and in periods of disease: dental carries, oral cavity cancer, periodontal diseases, cardiovascular diseases, diabetes mellitus, systemic sclerosis, and pancreatitis. The composition of saliva reflects dynamic changes in the organism, which makes it an excellent tool for determining clinically valuable biomarkers. The oral cavity and saliva may form the first line of defense against oxidative stress. Analysis of salivary antioxidants may be helpful as a diagnostic, prognostic, and therapeutic marker of not only oral, but also systemic health.

## 1. Introduction

Homeostasis is generally defined as a self-regulatory process in which biological systems maintain stability and, at the same time, adapt to changing external conditions. Of course, this concept also explains how the body can maintain constant internal conditions. It is an ability to adapt and survive in a changing and often hostile external environment [1]. In the 19th century, French scientist Claude Bernard was the first to articulate the hypothesis that the stability of the internal environment is independent of external conditions. He pointed out the importance of maintaining the ‘milieu intérieur’ [1,2]. Based on this idea of Claude Bernard, Walter Bradford Cannon introduced the term ‘homeostasis’ for the first time in 1926 [3].

Claude Bernard probably was the first to introduce the idea that every living system has its internal stability, protecting and buffering the body against the changing external environment. He also realized that the body has mechanisms that coordinate together to maintain a relatively constant temperature and concentration of blood glucose. This internal stability is vital for the organism’s health [1]. Homeostasis is a dynamic process in which internal conditions change in response to external challenges, resulting in the organism’s survival. Homeostatic regulation involves the complex interaction of several feedback systems, which higher control centers regulate. The final result of homeostatic regulation is the health and vitality of the organism [1]. The homeostasis at the cellular level is observed in ongoing biochemical reactions. For the optimal functioning of enzymes, pH regulation, temperature regulation, oxygen concentration, ion concentration, and blood glucose concentration are essential. The generation of waste substances is also constantly monitored to avoid disturbing the internal environment of the cells [4]. All body tissues form organs that are included in organ systems, which do not work independently, but must work together to achieve homeostasis. Every single cell in the body benefits from homeostatic control but also contributes to maintaining homeostasis, which provides the body with continuous automation [5].

Redox biology is a significant field aiming to explain and understand the living system [6]; the field’s role in health and disease is highly associated with the homeostasis of reactive oxygen species (ROS). Oxygen is considered the most common oxidizing agent, and oxygen reduction is essential for life in managing energy production during cell respiration. Also, the term oxidation itself describes the reduction of oxygen to water, and for targeted oxidation, the transformation of secondary chemical oxygen species [7]. Flohé et al. (2020) state in their work that “life science means redox science”, the origins of which are connected with the formation, metabolism, and functioning of reactive forms of oxygen and nitrogen in the body [8]. Redox biology discusses how redox homeostasis is regulated and maintained in the organism. It also tries to elucidate the disturbed mechanisms leading to the development of various pathological conditions caused by increasing oxidative stress [9]. Nowadays, the double effects of oxidative stress are well known. On the one hand, it positively affects physiology and health in redox signaling, known as “oxidative eustress”. On the other hand, increased oxidative stress causes damage to biomolecules, with a detrimental impact on pathophysiology [10]. Oxidative stress is linked with certain diseases; it plays a crucial role in the pathogenesis of some chronic diseases such as: diabetes mellitus, cardiovascular diseases (e.g., atherosclerosis and hypertension), cancer (e.g., colorectal, prostate, breast, lung, and bladder cancers), neurodegenerative diseases (e.g., Parkinson’s disease-PD, Alzheimer’s disease-AD, and Multiple sclerosis-MS), respiratory diseases (e.g., asthma), cataract development, and rheumatoid arthritis. Oxidative stress may cause alterations in the functioning of several enzymes and cellular structures, leading to abnormalities in gene expression. Oxidative stress is also associated with structural defects in mitochondrial DNA. [6,11,12,13]. Generally, oxidative stress is defined as an imbalance between the body’s free radicals and antioxidants [9]. Free radicals are reactive species derived from oxygen, nitrogen, or sulfur, containing an odd number of electrons. Free radical reactions in the human body can be beneficial or harmful to the body. Free radicals at physiological concentrations act as signaling molecules in signaling processes known as redox signaling and are essential in the intracellular destruction of bacteria by phagocytes, especially that by granulocytes and macrophages. At high concentrations, they cause damage to macromolecules in cells, initiating negative chain reactions causing cell membrane damage, inactivity of major enzymes, disruption of cellular processes important for the normal functioning of the cells, and inhibition of the normal division of the cells. High concentrations of free radicals cause damage to DNA, lipids, and protein. In the body, free radicals can be formed during numerous biological processes; for example, breathing, food digestion, metabolization of drugs and alcohol, or fat metabolism leading to energy production could all be their source. [11,12]. The activity of free radicals is inhibited and destroyed in the body due to the activity of the antioxidant system. Antioxidants are molecules that donate an electron to free radicals without becoming unstable [9]. Interactions and competitions between a network of enzymes, reactive species, and small antioxidants are an essential part of a complex homeostatic system, the loss of which leads to disease. Early therapy is very effective and must be aimed at maintaining or restoring homeostatic conditions [1,2].

## 2. Cellular Reactive Species

Reactive species include free radicals and reactive molecules generated in the cells by various metabolic reactions. Modern medicine has observed their dual effect upon the body. On the one hand, if these compounds are found in low concentrations in normal cells, they show beneficial effects, e.g., their importance for signal transduction. On the other hand, the high production of reactive species is associated with damage to tissue, followed by the development of diseases. Reactive species are divided into four groups; the first one is reactive oxygen species (ROS), the second one reactive nitrogen species (RNS), the third one reactive sulfur species (RSS), and the fourth one reactive carbonyl species (RCS) (Figure 1).

### 2.1. Reactive Oxygen Species (ROS)

The theory of oxygen free radicals has been known for more than 50 years. In the last two decades, their role in the development of diseases has been discovered [14]. ROS represents a group of highly reactive molecules derived from oxygen. ROS can be divided into radical and non-radical (molecules or ions) oxidants. To the ROS belongs superoxide anion (^•^O_2_^−^), hydroxyl radical (OH^•^), singlet oxygen (^1^O_2_), peroxyl radical (ROO), ozone (O_3_), hydrogen peroxide (H_2_O_2_), nitric oxide (^•^NO), peroxynitrite anion (ONOO^−^), and hypochlorous (-bromous) acid (HOCl, HOBr) [9].^•^NO and ONOO^−^ are technically reactive nitrogen species (RNS), but because RNS are almost exclusively oxygen-containing species, they may also be classified as ROS according to the definition given.

There are endogenous and exogenous sources of ROS. Within the cell, the leading producers of ROS are the electron transport chain and enzyme-catalyzed reactions involving NADPH oxidase, xanthine oxidase, nitric oxide synthase, cytochrome P450 enzymes, lipoxygenase, and cyclooxygenase [15]. Intracellular compartments involved in the formation of ROS are mainly mitochondria, peroxisomes, endoplasmic reticulum, nuclei, the cytosol, plasma, and membranes, while extracellular space can represent an alternative site of origin. ROS could also be the product of interaction with exogenous sources such as bacterial invasion [16,17]. Other cellular sources of ROS include neutrophils, monocytes, cardiomyocytes, and endothelial cells. ROS are also produced in metal-catalyzed free radical reactions [18]. The exogenous sources contributing to ROS formation are pollution, cigarette smoke, heavy metals, transition metals, industrial solvents, pesticides, insecticides, certain drugs like halothane and paracetamol, and UV radiation [13]. With relation to the oral cavity, ROS are produced during periodontal inflammation. Several beverages, including green and black teas and especially instant coffee, can contain H_2_O_2_; oral bacteria also produce H_2_O_2_ [19]. Other sources of ROS in the oral cavity might be xenobiotics (e.g., ethanol, cigarette smoke, drugs), food (e.g., high-fat diet, high-protein diet, acrolein), dental treatment (e.g., ozone, ultrasound, non-thermal plasma, laser light, ultraviolet light), and dental materials (e.g., fluorides, dental composites, fixed orthodontic appliances, and titanium fixations) [20].

#### 2.1.1. ROS Generation in the Human Body

The main source for the generation of ROS is oxygen. ROS originate in the reaction of oxygen’s reduction to water. This process involves the action of mitochondrial complexes, NADPH oxidases, peroxisomes, the microsomal electron transport chain, oxidase, cyclooxygenases, and xanthine oxidase [21]. Mitochondria are important in forming ROS, namely by means of the electron transport chain. In complexes I and II, the electrons are passed from reduced substrates through mitochondrial complexes III and IV to oxygen, generating water and the proton (H^+^) gradient required for energy production. If electrons leak before the reduction of O_2_ to water, direct one-electron reduction of O_2_ generates superoxide anion (^•^O_2_^−^). In mitochondria, complexes I and III produce the highest amount of superoxide [7,21].

In the plasma membrane, the superoxide anion is generated by the enzyme complex NOX and is used to kill pathogens. Superoxide anion is a very short-lived radical; it is converted spontaneously or enzymatically. Enzymes that catalyze superoxide conversion to more stable H_2_O_2_ are SODs. H_2_O_2_ (non-radical molecule) is mainly formed by the action of plasma membrane NADPH oxidases and by the reaction catalyzed by superoxide dismutase (dismutation of superoxide radical). H_2_O_2_ is also produced in peroxisomes by the peroxisomal enzymes like catalase (CAT) and glutathione peroxidase (GPx) and is decomposed here by catalase. Hydrogen peroxide is generally toxic to the cells. It has the ability to permeate through membranes in two ways, directly or by using aquaporin channels. The cells contain peroxidase enzymes catalyzing the reaction of H_2_O_2_ detoxification to water. In the cytoplasm, another hydrogen peroxide and superoxide source is the xanthine oxidoreductase complex [9,21]). Hydrogen peroxide can be converted to a highly reactive hydroxyl radical by exposure to UV light or by reaction with transition metal ions. The hydroxyl radical is the radical with the highest reactivity directly in the place of formation. Organisms contain mechanisms for sequestration of transition metal ions into protein-bound forms, preventing the catalysis of the OH radicals’ formation, which is essential in such extracellular fluids as blood plasma [19]. Singlet oxygen (^1^O_2_) represents the radical type of ROS; it is generated by photochemical activation and acts in the oxidation of LDL cholesterol, with effects on the cardiovascular system [9].

#### 2.1.2. ROS Effect on the Human Body

ROS show a double impact on the human body. At low levels, they are essential to physiological homeostasis, act as second messenger signaling molecules, and serve as defense mechanisms against invaders. They play a significant role in cellular processes such as cellular proliferation, differentiation, and migration, and activate stress-responsive survival pathways [9,22,23]. ROS affect cell-signaling proteins (NF-κB, MAPKs, Keap1-Nrf2-ARE, and PI3K-Akt), ion channels, and transporters (Ca^2+^ and mPTP), and the modifying protein kinase and ubiquitination/proteasome system [24]. From the ROS group, superoxide anion and hydrogen peroxide represent key molecules acting as cell biology signaling agents. The intracellular concentration of H_2_O_2_ is kept under control at low nanomolar levels (1–100 nM). The intracellular concentration of superoxide anion is much lower (10^–11^ vs. 10^–8^ M) compared to H_2_O_2_ [25]. In contrast to the physiological levels (for example, high concentrations of H_2_O_2_ are above 100 nM), ROS at high levels contribute to several kinds of cell damage and cell death [9,16]. Excessive amounts of all mentioned ROS result in oxidation of lipid, proteins, and DNA, and cause damage resulting in the altered structural and functional integrity of cells. [26,27]. ROS could damage proteins causing alteration in site-specific amino acids, fragmentation of the peptide chain, changes in electric charge, and inactivation of enzymes. ROS cause structural modification of cellular proteins, followed by cellular dysfunction and disturbance of vital cellular processes. Another biomolecule that ROS damage are lipids; ROS could break lipid membranes, causing increased membrane fluidity and permeability. ROS can also damage DNA molecules, or cause DNA methylation, deoxyribose oxidation, breaking of the DNA double strands, nucleotides removing and modifying bases, and intra- and interstrand cross-links. Oxidative damage of DNA leads to mutagenesis and cancer development [11,25,28,29,30].

### 2.2. Reactive Nitrogen Species (RNS)

To the group of RNS belong various nitric oxide-derived compounds, including nitric oxide (^•^NO), nitrous oxide (N_2_O), nitroxyl anion (NO^−^), nitrosonium cation (NO^+^), peroxynitrite (ONOO^−^), peroxynitrous acid (HNO_3_), S-nitrosothiols, and dinitrosyl iron complexes [31,32]. RNS play a vital role in the physiologic regulation of many living cells (e.g., cardiomyocytes, platelets, nervous cells, juxtaglomerular cells, and smooth muscle cells). They have a pleiotropic effect on cellular targets after posttranslational modifications and interactions with reactive oxygen species. High levels of RNS induce nitrosation stress and thus contribute to cell damage and death [32]. ^•^NO easily reacts with ^•^O_2_^−^ to form peroxynitrite (ONOO^−^), and this reaction depletes the biological activity of ^•^NO, thus affecting smooth muscle tone, platelet activation, blood pressure, and other signaling mechanisms [33]. Peroxynitrite is transformed into other reactive nitrogen species, which react and modify lipids, amino acids, nucleotides, and thiols and exerts various cytotoxic responses [34].

Nitrates and nitrites have long been assumed to be inert and have therefore been incorrectly used as an index of NOS activity, reflecting the degree of endothelial dysfunction in humans. Under physiological conditions, ^•^NO is eliminated to nitrates and nitrites via rapid oxidation via endogenous nitric oxide synthase (NOS), thus preventing ^•^NO accumulation and ONOO^−^ formation. In the oral cavity, nitrates, whose dietary intake is considerable, undergo a series of reduction reactions by the action of oral commensal bacteria, during which they are gradually reduced back to ^•^NO via nitrites [35]. For this reason, dietary interventions affecting the conversion of nitrate through nitrite to ^•^NO are considered particularly important in disease states where the NOS system is compromised [36,37,38]. However, the finding that the nitrate-nitrite-^•^NO pathway is gender-dependent is interesting. It was pointed out by Kapil et al. [39] that the oral reduction of nitrate is dependent on the oral microbiome, but also pointed out that females exhibited higher oral bacterial nitrate-reducing activity, although there were no differences between the genders in the composition of the total salivary microflora.

### 2.3. Reactive Sulphur Species (RSS)

Reactive sulfur species (RSS) represent a largely unknown group. Many results and observations in different studies pointed out the importance of RSS in redox homeostasis, cell signaling, and metabolism regulation [40]. The RSS group includes H_2_S, thiyl radical (HS^•^), hydrogen persulfide/polysulfide (H_2_S_n_; n ≥ 2), supersulfide radical (^•^S_2_ ^−^), protein thiols (PSH), small-molecular-weight thiols (RSH), small-molecular-weight thiol persulfide (RSSH), protein persulfide (PS-SH), various higher order polysulfides (RSS_(n)_H, RSS_(n)_R, and H_2_S_n_; n > 1; R = Cys, GSH, H), sulfenic acids (RSOH), nitrosothiols (RSNO), and various sulfide bridge forms (PS-SR, RS-S-SR, and RS-S_n_-SH) [9,41,42,43,44]. The central molecule of sulfur cell chemistry is gas H_2_S, which has the ability to pass through a biological membrane. It is formed at various places in the cytoplasm and the mitochondria (primarily from cysteine and homocysteine, but also from thiosulfate and carbonyl sulfide). H_2_S is inactivated in mitochondria by sequential oxidation to sulfate [9,43,44].

### 2.4. Reactive Carbonyl Species (RCS)

Reactive carbonyl species (RCS) are biological compounds with one or more carbonyl groups generated by non-enzymatic and enzymatic reactions. Dicarbonyls are more reactive than one carbonyl RCS; thus, despite a low concentration in tissue, still relevant damaging agents [45]. Oxidation of lipids, saccharides, and amino acids may provide more than 20 RCS [46,47]. Non-enzymatic processes leading to RCS generation are lipoperoxidation, glycation, and oxidation of amino acids; enzymatic pathways leading to RCS production are polyol pathway, alcohol oxidation, cyclooxygenase action, and ubiquitous glycolysis. RSC generated via non-enzymatic reactions include lipid hydroperoxides, malondialdehyde, 4-hydroxy-trans-2-nonenal, 4-oxo-trans-2-nonenal, glyoxal, methylglyoxal, acrolein, crotonaldehyde, hexanal, glucosone, 3-deoxyglucosone, glycolaldehyde, 2-hydroxypropanal, and isolevuglandins. Enzymatic sources give rise to the formation of 3-deoxyglucosone, 3-deoxyfructose, acetaldehyde, glyceraldehyde-3-phosphate, dihydroxyacetone phosphate, methylglyoxal and isolevuglandins [47,48,49]. Among those mentioned, glycation, lipoperoxidation, and alcohol oxidation products are of particular interest, as they are sufficiently potent to irreversibly modify macromolecules and form DNA, protein, and phospholipids adducts with a wide range of pathological effects. Interestingly, according to some studies, dietary antioxidants vitamin C and E are ineffective in preventing oxidative injury and lipid peroxidation, likely due to the essential physiological roles of reactive species [48].

### 2.5. Cellular Protection

Cells use small molecules called cellular antioxidants that prevent the accumulation of high levels of harmful free radicals. Antioxidants eliminate radicals and act against undesirable processes caused by oxidative stress [9]. Several groups of antioxidants are essential for homeostasis, for example: natural dietary antioxidants such as vitamins (A, C, E), endogenous antioxidant enzymes (catalase, superoxide dismutase, glutathione peroxidase, glutathione reductase, peroxiredoxins, ferritin with its ferroxidase activity) and small antioxidant molecules (coenzyme Q, glutathione, bilirubin) [16].

## 3. Saliva and Its Antioxidant Composition

In recent research, saliva has come to the forefront of scientific interest as an important biological fluid. It is a clear, viscous biological fluid. Saliva has many functions; it acts as a reservoir of ions, a buffer, and a cleansing solution, while providing local immunity, and modulating taste. Saliva has also antifungal, antibacterial, and antiviral activities [50]. It is secreted by the three pairs of major salivary glands. Saliva consists of 98% water, and the remaining 2% represents proteins, glycoproteins, electrolytes, small organic molecules, enzymes, hormones, antioxidants, and other substances [51,52].

Sampling of saliva is a safe, simple, and non-invasive procedure with a limited chance of infections, which is important, especially with children and teenagers, irritable patients, and patients who are disabled, either mentally or physically [50,53].

Saliva plays a crucial role in maintaining oral cavity health and physiology and provides the first line of antioxidant defense for gastrointestinal tracts [54]. An increasing number of studies suggest saliva analysis as an alternative to blood serum analysis for diagnostic purposes. The biomarkers found in saliva include antioxidant enzymes (for example, superoxide dismutase, catalase, glutathione peroxidase) and also products of oxidative damage (malondialdehyde, protein carbonyls, 8-hydroxydeoxyguanosine–8-OHdG, 4-hydroxyalkenals). Changes in salivary redox homeostasis may reflect the presence and seriousness of various oral and systemic conditions, for example: periodontal disease, diabetes, and inflammatory bowel disease [55].

## 4. Salivary Antioxidant System

Antioxidants are present in all body fluids, including saliva. Human saliva contains several antioxidants that serve to balance the effect of oxidants. Antioxidants present in saliva are both enzymatic (e.g., superoxide dismutase, catalase, peroxidase) and non-enzymatic (e.g., ascorbic acid, albumin, glutathione, lactoferrin, vitamins, and uric acid) [56]. Saliva plays an important role in the protection against oxidative stress. The oral cavity and saliva may form the first line of defense against oxidative stress, thereby providing protective effects against microorganisms, toxins, and oxidants [54,57,58].

### 4.1. Superoxid Dismutases (SODs)

SODs are in saliva as SOD1-copper-zinc superoxide dismutase, SOD2-manganese superoxide dismutase, and SOD3-extracellular superoxide dismutase [56]. SOD catalyzes the dismutation reaction of superoxide anion into hydrogen peroxide and molecular oxygen. Superoxide dismutase is an antioxidant enzyme that catalyzes two superoxide anion molecules into hydrogen peroxide (H_2_O_2_) and oxygen (O_2_) molecules [56,59]. Salivary SOD1 consists of two identical subunits; each contains one atom of copper and one atom of zinc. It is thermostable and resistant to proteolytic enzymes. SOD2 is found in the mitochondrial matrix, extracellular space, and peroxisomes. SOD3 is located in the extracellular space; it is a glycosylated protein, a hydrophobic enzyme with a molecular weight of 135 kDa, present in the form of di- and tetramer. SOD3 also has peroxidase activity. Its important function is to protect the biological activity of ^•^NO and prevent the formation of superoxide nitrate ion due to the dismutation reaction of superoxide anion [60].

### 4.2. Catalase

Catalase is a tetrameric protein with four similar subunits, each containing a heme group in the active site. The molecular weight of CAT is 240 kDa. CAT is an antioxidant enzyme that is present almost in all oxygen-using living organisms. It is located in the peroxisomes, mitochondria, endoplasmic reticulum, and cytosol. CAT uses iron or manganese as a cofactor and catalyzes the degradation of hydrogen peroxide (H_2_O_2_) produced by SOD activity to water and molecular oxygen [56,61].

### 4.3. Glutathione Peroxidase (GPx)

GPx breaks down H_2_O_2_ into water and lipid peroxides into their relevant alcohols. It plays an essential role in neutralizing hydrogen peroxide (H_2_O_2_) produced through the SOD dismutation process. It is mainly found in mitochondria, and sometimes in the cytosol [58].

### 4.4. Salivary Peroxidase and Myeloperoxidase

In the oral cavity, there are two main peroxidases: salivary peroxidase (sialoperoxidase, 80%) and myeloperoxidase (20%). Both enzymes represent innate host defense in the oral cavity. Peroxidase is a heme-containing enzyme that belongs to the oxidoreductase enzymatic group. Peroxidase reacts with hydrogen peroxide and organic peroxides to protect the body from oxidative stress [62]. The parotid and submaxillary salivary glands secrete salivary peroxidase. Salivary peroxidase is an enzyme whose active center contains selenium. It is present in salivary secretions and on the oral surfaces’ biofilms. It occurs in various forms (based on electrophoretic migration and molecular weight), but when salivary peroxidase is unbound, its molecular weight is approximately 78,000, and the isoelectric point is 8–10. In the oral cavity, it adsorbs different oral surfaces, such as enamel, salivary sediments, dental plaque, and bacteria, and at the same time retains its enzymatic activity [52,56,63]. Myeloperoxidase in the saliva is produced by the neutrophil’s lysis, where they undergo hypoosmotic shock and release myeloperoxidase together with other antimicrobial proteins-lysozyme and lactoferrin. Myeloperoxidase is abundant in the dento-gingival sulcus area [63].

The primary function of salivary peroxidase and myeloperoxidase in the oral cavity is the oxidation of thiocyanate ion (SCN^-^), which is derived from diet to hypothiocyanite (OSCN^-^) with antibacterial activity, in the presence of hydrogen peroxide H_2_O_2_ which is produced by oral bacteria and cells [19,52,56,62,63,64]. Furthermore, peroxidases present in human saliva consume bacteria-produced H_2_O_2_ and could help inactivate some toxic carcinogenic, mutagenic, and genotoxic substances [63,64].

## 5. Non Enzymatic Salivary Antioxidants

In addition to antioxidant enzymes, saliva is also rich in non-enzymatic antioxidants. The non-enzymatic salivary antioxidants are: uric acid, glutathione, albumin, lactoferrin, ascorbic acid (vitamin C), and vitamins A and E. The most crucial representants of this group are uric acid, ascorbic acid, and reduced glutathione [52,56]. The correlation between the level of uric acid in saliva and the level in the plasma proves that uric acid comes from plasma [65]. Uric acid (UA) represents more than 85% of the total antioxidant capacity of unstimulated and stimulated human saliva [51]. Glutathione (GSH) is a low molecular weight thiol involved as a coenzyme in oxidation-reduction reactions in cells. It is a tripeptide composed of three amino acids: cysteine, glutamate, and glycine [66]. Ascorbic acid (vitamin C) is an essential micronutrient for humans. Not long after its discovery, vitamin C was found in saliva. In 1935, vitamin C was measured in saliva at a concentration of 2.5 μg/mL [67].

## 6. Salivary Antioxidant Activity in Human Health and Disease

In recent decades, the antioxidant protection of saliva has been the subject of much biomedical research (Table 1). Saliva, as a diagnostic tool, comes to the forefront of scientific interest because saliva reflects the body’s levels of various hormonal, immunological, toxicological, and infectious disease markers, and it seems to be an excellent tool for monitoring not only oral but also systemic health [52].

## 7. Dental Caries

Dental caries is one of the most common health problems of the oral cavity, and its prevention is one of the most important strategies in many countries. Dental caries, like many other pathological conditions, can be related to increased reactive species formation. Saliva represents the first line of defense against dental caries and plays a vital role in preventing oral infection and caries.

Salivary malondialdehyde, a biomarker of cell membrane damage, is higher in both children [70] and adults [71] with caries, suggesting a link between lipid peroxidation and the caries process. The relationship between salivary total antioxidant capacity (TAC) and dental caries was evaluated in the study of Ahmadi-Motamayel et al. (2013). One hundred healthy high school students (50 women and 50 men) aged 15–17 took part in the study. They observed that the level of TAC was significantly higher in the saliva of the group with active caries compared to the subjects without caries. When comparing men and women, they observed statistically significantly lower levels of TAC in the group of women compared to men [68]. Similar results have been observed in other studies that repeatedly confirm an increase in TAC levels in persons with active caries. Their study showed similar results as results of other studies. Hegde et al. (2009) in their study investigated the TAC of saliva and its relation to early childhood caries and rampant caries. Their results indicated that TAC of saliva increased in children with caries, in addition, they reported that TAC of saliva increased with age in children [57]. In the study of Tulunglu et al. (2006) was observed, that the total protein and TAC of saliva increased with caries activity [91]. The study of Kumar et al. (2011) reported significant evaluation of TAC of saliva in children with severe early childhood caries [92].

Hendi et al. (2020) studied salivary superoxide dismutase, peroxidase, glutathione peroxidase, catalase, and uric acid in dental caries. Their study included 100 students, both male and female. They observed higher levels of peroxidase, uric acid, catalase, and glutathione peroxidase and lower levels of superoxide dismutase in the group with active caries compared to those without caries. When comparing the group of women and men, they observed lower levels of peroxidase, catalase, and uric acid and higher levels of glutathione peroxidase in the group of women [69]. Vahabzadeh et al. (2020) evaluated the correlation of salivary enzymatic antioxidant activity of SOD, CAT, and GPx with different levels of dental caries in children (7–12 years). Ninety healthy children (7–12 years; 36 girls and 54 boys) participated in their study. Although their results showed no significant correlation between SOD, CAT, or GPx and decayed-missing-filled teeth (DMFT) index for permanent/primary dentition, they observed changes in antioxidant enzymes related to dental caries. In proportion to the number of decayed teeth was increased CAT activity. Moreover, a positive correlation between SOD activity and tooth brushing frequency was found. When comparing girls and boys, higher activity of SOD, CAT, and GPx was observed in boys than in girls, and an inverse relationship between enzymatic antioxidant activity and age was detected [58].

## 8. Oral Cavity Cancer

The most common malignant tumor of the head and neck is squamous cell carcinoma of the oral cavity. The worldwide incidence of this disease is more than 300,000 new cases each year [93]. Many studies indicate a direct effect of cigarette smoke on the formation of tumors due to the attack of various compounds of cigarette smoke on the cells of the oral epithelium, which gradually accumulate and cause a gradual malignant transformation. This progressive process, which initially manifests as dysplastic mucosal lesions are induced by free radicals, reactive oxygen species, and reactive nitrogen species found in inhaled cigarette smoke. Mucosal lesions then transform into in situ carcinoma lesions and can lead to complete infiltration and metastasis of oral squamous cell carcinoma [94]. One source of free radicals is smoking. Since saliva is the first body fluid that cigarette smoke encounters when inhaled, it is important to know the level of salivary antioxidants to maintain a balance between antioxidants and free radicals. It can also help reveal the level of oxidative stress caused by cigarette smoke. Many studies have shown that patients with oral cancer showed lower total antioxidant capacity, uric acid concentration, salivary peroxidase, and superoxide dismutase activity in saliva compared to healthy subjects. This lesser capacity could be a risk factor for tumor induction [73].

Abdolsamadi et al. (2011) compared salivary antioxidant levels between healthy smokers and non-smokers [72]. Salivary uric acid levels, superoxide dismutase, glutathione peroxidase, and peroxidase activities were determined in their study, which included 80 men (40 smokers with daily consumption of 20 cigarettes for at least ten years and 40 non-smokers). They observed that mean levels of salivary superoxide dismutase, glutathione peroxidase, and peroxidase were significantly lower in smokers than in non-smokers but also observed no statistically significant difference in salivary uric acid levels between smokers and non-smokers. Baharvand et al. (2010) compared SOD levels in smokers and non-smokers. Sixty volunteers (30 smokers and 30 non-smokers) participated in the study. Their results showed that the level of SOD is significantly higher in smokers than in non-smokers due to the reduction of free radicals produced by smoking [59]. Hershkovich et al. (2007) investigated age-related changes in salivary antioxidants, as the prevalence of oral cancer is higher in the elderly. They found a significantly reduced total value of the antioxidant capacity of saliva in the elderly, increased oxidative stress, and increased concentrations in saliva and total RNS values. All of the above factors contribute to increased DNA oxidation of oral epithelial cells, which may contribute to a higher prevalence of oral cancer in the elderly population [95]. Singh et al. (2014) compared the level of salivary antioxidants uric acid (UA), glutathione S transferase (GST), and superoxide dismutase (SOD) between healthy control and a study group (patients with oral squamous cell carcinoma). Both groups consisted of 50 participants. They observed lower levels of UA and SOD in saliva in oral squamous cell carcinoma than in healthy patients, except for GST levels, which showed statistically significant increased levels compared to the healthy control group. They also observed a progressive increase (without statistical significance) in salivary SOD levels between well- and poorly-differentiated squamous cell carcinoma. The authors suggest that antioxidants may play a significant role in preventing oral cancer [74]. A huge meta-analysis of MDA levels also points to oxidative damage in oral cancer. MDA in saliva and plasma is increased in oral squamous cell carcinoma, but in tissue samples, MDA is attenuated [75]. Thus, understanding the pro- and antioxidant balance is essential for future therapeutic and detection strategies in cancer treatment.

## 9. Periodontal Diseases

Periodontal diseases belong to the group of the most widespread chronic diseases suffered by adults. Periodontal diseases can be divided into gingivitis and periodontitis [96]. Generally, it is believed that a microbial interaction initiates periodontal diseases. The first line of defense in oral tissue against pathogenic microorganisms is polymorphonuclear leukocytes (PMNs). When PMNs are activated, they produce large amounts of reactive oxygen species (ROS), causing oxidative stress, one of the main pathological patterns that cause periodontal tissue destruction [76,77]. RCS are also associated with periodontal damage. Significantly higher levels of salivary MDA and 8-hydroxydeoxyguanosine (8-OHdG), a biomarker of oxidative stress, were found in patients with chronic periodontitis. After non-surgical treatment, both markers decreased; 8-OHdG decreased significantly, and MDA insignificantly [79]. In the case of comorbidity with diabetes, salivary 8-OHdG, 4-hydroxynonenal, AGE, and AGE receptor showed the best correlations with the disease [80]. Syahputra et al. (2018) evaluated differences in salivary levels of SOD in patients with gingivitis and periodontitis. Their study involved 44 patients, of which there were 22 with gingivitis and 22 with periodontitis recruited from the periodontal installation. Their results showed a significant difference between salivary SOD levels of gingivitis patients and periodontitis patients. They observed higher levels of salivary SOD levels in patients with gingivitis than for patients with periodontitis [75]. Novakovic et al. (2014) investigated the effect of non-surgical periodontal treatment on salivary antioxidants. They also evaluated the capacity of salivary antioxidants as biomarkers that reflect the periodontal tissue condition and the therapy outcome. Their study included 63 systemically healthy non-smokers, of which 21 were periodontally healthy subjects, and 42 were patients with current chronic periodontitis. Half of the patients received scaling, and root planing and the other half received only oral hygiene instructions. All measured antioxidants were affected by treatment, and they observed a significant increase in TAC, ALB, UA, and GPX levels and a decrease in SOD levels in response to scaling and root planing, while no differences were observed for any of the parameters in the oral hygiene instructions group. The authors concluded that non-surgical periodontal treatment affected salivary TAC, ALB, UA, SOD, and GPX. These biochemical parameters convincingly reflected the condition of the periodontium and the tissue’s response to treatment [77].

## 10. Cardiovascular Diseases (CVDs)

CVD is a general term for disorders affecting the heart and blood vessels. Oxidative stress has been shown to play an essential role in the pathogenesis of cardiovascular disease. There is an association between periodontitis and cardiovascular disease [97]. CVD and periodontitis may have common risk factors for their occurrence, such as smoking, dietary habits, socio-economic status, diabetes, and free radicals [78]. A study of Dhotre et al. (2011) was conducted to assess the possible mechanisms underlying the pathogenesis of both periodontitis and cardiovascular diseases. Their study included 100 patients with periodontitis and 100 healthy controls. They measured periodontal status, serum and salivary antioxidants, and total antioxidant capacity; they also screened the total lipid profile as a CVD risk marker. Based on their results, the authors concluded that increased oxidative stress and altered lipid profile in patients with periodontitis could contribute to the development of cardiovascular disease in these patients. They observed a significant increase in periodontal depth and clinical attachment loss in patients with periodontitis compared to healthy controls [78].

The latest research shows that the step that connects oral infection with vascular diseases is the subsequent activation of the host’s inflammatory response after the bacteria or their products enter the bloodstream. It occurs through several mechanisms that support the formation, maturation, and exacerbation of atheromatous lesions. C-reactive protein, matrix metalloproteinases, fibrinogen, and other hemostatic factors are among the most prominent inflammatory mediators inducing the formation and progression of atheroma, mainly through oxidative stress, lipid oxidation, and inflammatory dysfunction [97].

A study by Pussinen et al. in which participants from age six at baseline were followed for 27 years suggests that oral infections in childhood are associated with subclinical carotid atherosclerosis in adulthood, and vice versa, periodontal treatment improves the atherosclerotic profile by reversing endothelial dysfunction and reducing inflammatory and lipid biomarkers [98].Several studies on chronic heart failure (HF) suggest that oxidative stress plays an important role in its pathogenesis of the disease, a leading cause of adult death. The secretory dysfunction of the salivary glands may also occur in patients with HF. Klimiuk et al. (2020) evaluated the relationship between salivary oxidative stress and HF. In their study, they compared the rate of ROS generation, enzymatic and non-enzymatic antioxidant barriers, and oxidative damage to proteins and lipids in unstimulated and stimulated saliva, as well as plasma/erythrocytes of patients with chronic heart failure and healthy controls. The study included 50 patients with HF. The authors were the first to prove that the patients with the progression of HF had enzymatic and non-enzymatic antioxidant defense disorders. They observed oxidative damage of proteins and lipids at both the central level (plasma/erythrocytes) and the local level (saliva). They observed decreased salivary secretion, decreased level of total salivary protein, and decreased salivary amylase activity. Their results indicate secretory dysfunction of the salivary glands in patients with HF compared to an age- and gender-matched control group. In patients with HF, the submandibular salivary glands are mainly affected. Disorders of redox homeostasis generally worsen with the progression of HF [81].

Some oxidative stress parameters in saliva can be potential diagnostic biomarkers of CVDs. Punj et al. (2017) investigated the relationship between periodontitis and coronary heart disease in 80 individuals, patients aged 25 to 65 [82]. They determined total antioxidant capacity, superoxide dismutase, glutathione peroxidase, and catalase in saliva and serum. Their results showed a statistically significant decrease in total antioxidant capacity, superoxide dismutase, catalase, and glutathione peroxidase in patients with chronic periodontitis and ischemic heart disease with or without periodontitis compared to healthy patients. They observed no uniformity in the correlation of serum and salivary antioxidant levels, raising the question that saliva may not serve as a surrogate for a serum for the diagnostic purpose of antioxidant determination [82]. Narain et al. (2012) determined salivary and serum antioxidant activity in patients with the acute coronary syndrome (ACS). They evaluated the redox status of patients with ACS before and after treatment compared to age-matched healthy individuals. In their study, they determined the antioxidant potential of serum and saliva. Their study included 33 patients (men aged 29–78) diagnosed with ACS and 16 healthy men (control). They observed a significantly attenuated antioxidant defense in patients with ACS compared to the control group, indicating increased oxidative stress. Their results pointed out that ACS patients who received long-term drug treatment (4–5 weeks) showed a significant increase in antioxidant activity and improved their clinical condition. Their study supports the use of the antioxidant activity of saliva and serum as a potential marker of redox status [83].

## 11. Diabetes Mellitus

Diabetes mellitus (DM) is a severe disease, a progressive metabolic disorder that affects many people worldwide. It is estimated that by 2025, the number of people affected by DM will increase to 300 million worldwide. DM development is related to several mechanisms, including oxidative stress. Increased oxidative stress is an accepted contributor to the development and progression of DM and its complications [99]. As a precursor to diabetes, even insulin resistance is related to a significant depletion of antioxidant protection and increased protein and lipid oxidation products. Salivary 4-HNE and GSH significantly correlate with body weight parameters, such as body weight and BMI, but also with biochemical markers of impaired carbohydrate metabolism and some pro-inflammatory adipokines leptin, resistin, and TNF-α in rats with insulin resistance [86]. These results could highlight salivary GSH and 4-HNE as possible biomarkers of insulin resistance progression. Fathi et al. (2020) investigated the relationship between antioxidants and markers of oxidative stress in the saliva of patients with DM type 2 (DM2) and a healthy control group. 20 patients with diabetes and 20 healthy subjects participated in their study. The authors determined salivary antioxidant markers (total antioxidant capacity (TAC), uric acid (UA), peroxidase, and catalase) and markers of oxidative stress (total oxidation state (TOS), malondialdehyde (MDA), and total thiols (SH)). The authors observed a significant decrease in TAC and catalase in DM2 patients. Also, salivary UA and peroxidase levels were non-significantly lower in DM2 patients. Of oxidative stress markers, they observed significantly higher MDA and TOS levels in DM2 and only slightly higher SH levels in DM2 patients [84]. Arana et al. (2017) determined differences in salivary oxidative stress between patients with type 2 diabetes mellitus (DM2) and healthy controls. They investigated the effect of oxidative stress on periodontal disease in patients with diabetes. Their study included 70 probands divided into three groups (19 patients without diabetes (control group), 24 patients with good metabolic control, and 27 DM2 patients with poor metabolic control. They determined the levels of glutathione peroxidase (GPx), glutathione reductase (GRd), reduced glutathione (GSH), and oxidized glutathione (GSSG). The authors observed a significant increase in the activity of GPx and GRd in the group of people with diabetes with good metabolic control compared to the healthy control group and a significant decrease in the activity of the measured enzymes in patients with poor metabolic control compared to the control group and well controlled diabetic groups. They also observed higher GSSG/GSH quotients in both diabetic groups. Their results suggest that salivary oxidative stress in DM patients is also associated with worse periodontal health [85]. Ahmadi-Motamayel et al. (2021) evaluated salivary antioxidants and oxidative stress in gestational diabetes mellitus (GDM), a common pregnancy complication. Their study included 20 pregnant women with GDM and 20 healthy pregnant women. The authors observed reduced levels of all antioxidant markers (uric acid, total antioxidant, peroxidase, and catalase) in the GDM group. They also observed increased markers of oxidative stress (salivary malondialdehyde, total oxidative stress, and total thiols) in the GDM group. Levels of catalase and total oxidative stress may be important in showing early changes in GDM [87]. It is evident that the pathophysiological mechanisms of DM are coupled to increased oxidative stress and reduced antioxidant capacity levels. Another aspect is the increased NO production during the early phases of nephropathy in DM, which swings to a suppressed NO bioavailability in the kidneys in the later strage, up to a state of generalized NO deficiency in severe diabetes with profound insulinopenia [100]. Dietary nitrate supplementation contributes to nitrate-nitrite-^•^NO pathway, has a direct effect on mitochondrial respiration and thus the reactive oxygen species homeostasis. It has also been associated with the activation of AMP-activated protein kinase (AMPK), improving glucose homeostasis and insulin sensitivity [101]. Schiffer et al. (2020) proposed an ^•^NO-independent nitrite augmented mitochondrial fusion-dependent glucose uptake in adipocytes by activating protein kinase A (PKA) and protein kinase G (PKG). In their study, among others, they emphasized the role of nitrate supplementation in DM in browning of the white adipose tissue, insulin, and ^•^NO-independent GLUT4 translocation and attenuating oxidative stress via reduced NADPH oxidase activity [102].

## 12. Systemic Sclerosis

Systemic sclerosis (SS) is an autoimmune disease that causes the atypical growth of connective tissues. Oxidative stress is believed to be involved in the pathogenesis of this disease. There is little evidence to analyze the antioxidant profile of saliva in such patients, however, one of the oral manifestation of patients with progressive SS is dry mouth and parodontopathy [102]. Su et al. (2010) evaluated salivary redox homeostasis in patients with SS in 190 probands (70 women with SS and 120 control women). Levels of 8-OHdG, 8-epi-prostaglandin F2α (8-epi-PGF2α), and total protein carbonyls were analyzed. Based on the results, they concluded that salivary redox homeostasis is disturbed in patients with SS and can inform about the pathophysiology, the presence of the disease, and the effectiveness of therapeutic interventions [55]. Zalewska et al. (2014) also studied salivary antioxidants in patients with SS. They evaluated unstimulated and stimulated salivary flow, peroxidase and SOD activity, total UA level, and total antioxidant status in a group of SS women and a healthy control group. They observed a significant decrease in the specific activity of peroxidase in unstimulated and a significant decrease in all investigated antioxidants in stimulated saliva in the group with hyposalivation compared to the group with normal saliva flow. Their results show that damage to the salivary glands during systemic sclerosis can be attributed to free radicals and correlates with the duration of the disease [88]. The topic of salivary biomarkers in SS disease is addressed in the work of Zian et al. (2018), who summarized the small number of available studies on the proteins present in the saliva of SS patients that could be used in the early disease detection and prognosis of the disease [103]. Representatives of such proteins include S-100 proteins (S-100A7-psoriasin, A8–calgranulin-A; A9–calgranulin-B); however, they are not highly specific since they have been identified as key players in the pathogenesis of other diseases, such as DM, psoriasis, and various types of cancer [104]. An exciting representative highlighted as a possible salivary SS disease marker is glyceraldehyde 3-phosphate dehydrogenase (GAPDH), one of the key glycolytic enzymes. GAPDH is one of the most prominent cellular targets of post-translational modifications (PTMs), especially during oxidative stress, which diverts GAPDH towards various non-glycolytic functions, e.g., maintaining cellular iron homeostasis [105,106]. Various oxidative post-translational modifications of GAPDH include, e.g., sulfenylation, S-thiolation, nitrosylation, and sulfhydration [106]. These modifications alter the enzyme’s conformation, subcellular localization, and regulatory interactions with downstream metabolites, affecting both its glycolytic and non-glycolytic functions. Oxidation-induced dysregulation of GAPDH results not only in autoimmune but also in both neurodegenerative and metabolic disorders.

## 13. Pancreatitis

Acute pancreatitis (AP) is a multifactorial disease characterized by pancreatic necro-inflammatory changes related to the dangerous release of digestive enzymes and pro-inflammatory cytokines into the pancreatic interstitium, surroundings, and the general circulation. The incidence of AP ranges from 10–80,000 to 100,000 annually, and the overall mortality of hospitalized patients is dramatic at around 10% [54,107]. The pathogenesis of acute pancreatitis is still not fully understood, and it is assumed that oxidative stress is one of the factors involved in its development [108]. The pancreas and salivary glands have been shown to share many histological and functional similarities. Autoimmune pathogenesis has been observed in some cases of chronic pancreatitis. Kamisawa et al. (2003) evaluated the frequency of salivary gland dysfunction in patients with chronic pancreatitis of various etiologies [90]. Patients with chronic pancreatitis or Sjogren’s syndrome and various control groups participated in their study. The etiologies of chronic pancreatitis were alcoholic, idiopathic, and autoimmune. The authors observed that salivary glands’ function was impaired in chronic pancreatitis of various etiologies. Salivary gland dysfunction may result from the pathophysiological effects of alcohol in patients with alcoholic chronic pancreatitis and an aggressive immune mechanism against the pancreas and salivary ducts in patients with autoimmune and idiopathic chronic pancreatitis [90]. Maciejczyk et al. (2019) evaluated the relationship between free radical production, enzymatic and non-enzymatic antioxidants, oxidative damage, and secretory function of the salivary glands of male rats with AP. The authors demonstrated that superoxide dismutase and glutathione reductase activities and glutathione concentration were significantly reduced in both the parotid and submandibular glands of AP rats compared to control rats. They concluded that AP increases the production of free oxygen radicals, worsens the redox balance of salivary glands, and is responsible for more serious oxidative damage to these glands. Oxidative modification of proteins and dysfunction of the antioxidant barrier is more pronounced in the submandibular glands [54].

## 14. Aging

The link between oxidative stress and aging is undoubted. Salivary biomarkers could represent a non-invasive alternative for aging measurement. Along with many other possible biomarkers of oxidation stress, salivary GSSG is age-linked [89]. AGE and 8-OHdG showed excellent diagnostic utility in assessing the aging process, helping to differentiate children and adolescents up to 15 years of age from the middle-age population and people over 65 years. Salivary AGE and 8-OHdG levels also reflect their plasma levels, contrary to many other oxidative damage markers that change with age, e.g., total thiols, carbonyl groups, carboxymethyl-lysine, carbamyl-lysine, and 4-hydroxynonenal, but correlating poorly with plasma levels if aging is taken into account [109].

## 15. Conclusions

Saliva is a biological fluid with extensive diagnostic use. Scientific interest in using this biological fluid is constantly growing due to its safe, simple, and non-invasive sampling. The composition of saliva reflects dynamic changes in the organism, which makes it an excellent tool for determining clinically valuable biomarkers. Since changes in salivary redox homeostasis may reflect the presence and severity of various oral and systemic diseases, analysis of salivary antioxidants may be helpful as a diagnostic, prognostic, and therapeutic marker. A growing number of studies suggest saliva analysis as an alternative to blood serum analysis for diagnostic purposes. Further prospective, interventional, and experimental studies using sufficient saliva samples and detailed selection criteria are needed to obtain as many results as possible in order to evaluate whether the diagnostic potential of saliva is comparable to that of blood serum.

## Figures and Tables

**Figure 1 ijms-23-10076-f001:**
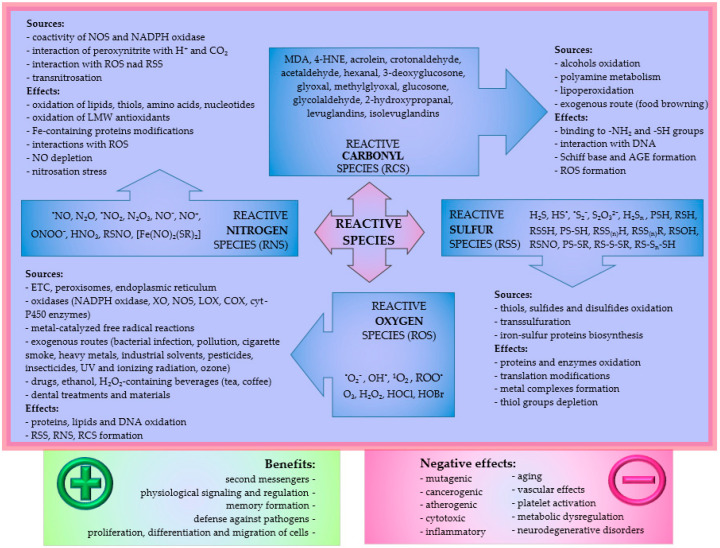
Reactive species, their sources and effects. **Legend:** NO-nitric oxide, NOS-nitric oxide synthase, NADPH-oxidase-nicotinamide adenine dinucleotide phosphate oxidase, LMW-low molecular weight, XO-xanthine oxidase, LOX-lipooxygenase, COX-cyclooxygenase, AGE-advanced glycation end products, MDA-malondialdehyde, 4-HNE-4-hydroxynonenal.

**Table 1 ijms-23-10076-t001:** Salivary redox markers in various pathological states.

State	Marker	Finding	References	Notes
**DC**	TAC	↑	[57,68]	
	SOD	↓	[58,69]	
	Px	↑	[69]	
	UA	↑		
	CAT	↑	[58,69]	
	GPx	↑		
	MDA	↑ ↑	[70] [71]	Early childhood caries Adults
**SM**	Px	↓	[72]	
	SOD	↓ ↑	[72] [59]	
	GPx	↓	[72]	
**OC**	UA	↓	[73,74]	
	Px	↓	[73]	
	SOD	↓	[74]	
	GST	↑		
	MDA	↑	[75]	Data from meta-analysis
**PO**	SOD	↓	[76,77]	After treatment
	TAC	↑ ↓	[77] [78]	
	ALB	↑	[77]	
	UA	↑		
	GPx	↑		
	MDA	↑	[78,79,80]	Treatment does not improve levels significantly
	NO	↑	[78]	
	8-OHdG	↑	[79,80]	Improved levels after treatment, also in diabetics
	4-HNE	↑		In diabetes
	AGE	↑		
	RAGE	↑		
**HF**	TP	↓	[81]	Mainly submandibular salivary glands
	A	↓		
	SOD	↑		
	CAT	↑		
	Px	↓		
	GSH	↓		
	UA	↑		
	TAC	↓		
	TOS	↑		
	TAC/TOS ratio	↑		
	ROSP	↑		
	AGE	↑		Correlation with AGE plasma levels, serum NT-proBNP Negative correlation with cardiac ejection fraction
	MDA	↑		Correlation with plasma levels
	AOPP	↑		
**CHD**	TAC	↓	[82]	No correlation to serum levels
	SOD	↓		
	CAT	↓		
	GPx	↓		
**ACS**	AA	↓	[83]	More significant in STEMI, improvement in antioxidant activity after 6–5 weeks simultaneously with clinical status
**DM2**	TAC	↓	[84]	
	CAT	↓		
	TOS	↑		
	MDA	↑		
	GPx	↑ ↓	[85]	Good metabolic control Poor metabolic control
	GR	↑ ↓		Good metabolic control Poor metabolic control
	GSSG/GSH	↑		
**IR**	SOD	↓	[86]	In rats, correlation with blood levels
	ASCA	↓		
	GSH	↓		In rats, correlation with blood levels, body weight, BMI, glucose, insulin, HOMA-IR, leptin, resistin and TNF-α
	PC	↑		In rats, correlation with blood levels
	4-HNE	↑		In rats, correlation with blood levels, body weight, BMI, glucose, insulin, HOMA-IR, leptin, resistin and TNF-α
	AGE	↑		In rats, correlation with blood levels
	3-NT	↑		
**GDM**	UA	↓	[87]	
	TAS	↓		
	Px	↓		
	CAT	↓		
	MDA	↑		
	TOS	↑		
	TT	↑		
**SS**	log PC	↑	[55]	Non-significant enhanced protein oxidation without concomitant lipid peroxidation and DNA damage
	Px	↓	[88]	In hyposalivation
	SOD	↓		
	UA	↓		
	TAS	↓		
**Aging**	AGE	↑	[86]	Correlates with plasma levels
	8-OHdG	↑		
	CML	↑		
	CL	↑		
	TT	↓		
	PC	↑		
	4-HNE	↑		
	GSSG	↓	[89]	
**PCT**	β2MG	↑	[90]	
	SOD	↓	[54]	In rats, from salivary gland homogenates
	GR	↓		
	GSH	↓		
	A	↑		

TAC–total antioxidant capacity, TAS–total antioxidant status, SOD–superoxid dismutase, TT–total thiols, TOS–total oxidative stress, UA–uric acid, MDA–malondialdehyde, Px–peroxidase, CAT–catalase, GPx–glutathion peroxidase, GR–glutathion reductase, GSSG–oxidized glutathion, GSH–reduced glutathion, β2MG-β2-microglobulin, PC–protein carbonyls, A–amylase, AA–antioxidant activity, AOPP-advanced oxidation protein products, AGE-advanced glycation end products, ASCA–ascorbic acid, ROSP–ROS production, TP–total protein, NO-nitric oxide, 8-OHdG–8-hydroxydeoxyguanosine, 4-HNE–4-hydroxynonenal, 3-NT–3-nitrotyrosine, CML–carboxymethyl-lysine, CL–carbamyl-lysine, ALB–albumin, GST–glutathion S transferase, SS–Systemic sclerosis, PCT–pancreatitis, DM2–diabetes mellitus type 2, IR–insulin resistence, GDM–gestational diabetes mellitus, ACS-acute coronary syndrome, CHD-coronary heart disease, HF-heart failure, PO–periodontitis, OC-oral cancer, SM–smoking, DC-dental caries.

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
