# Peer review of "Salivary Redox Homeostasis in Human Health and Disease"

_ijms, 2022, doi:10.3390/ijms231710076_

Round 1
Reviewer 1 Report
This review is generally well written but there are some issues that have been identified as follows:
1. Some of the text herein is superficial and there needs to be much more balanced description that includes more specific details. The field of redox biology is littered with reviews that fail to provide sufficicent detail on specific ROS, how they are generated and what characteristic damage they can cause in biological systems leaving researchers to accumulate all oxidants under a single banner ROS to simplify things. However, redox regulation is linked to the species being generated and there is specific pathways being elicited and specific damage being caused by different reactive species.
2. Reference to oxidative or reducing stress: This terminology is not used commonly - it is more considered to write increasing oxidative stress and/or limiting endogenous antioxidant capacity thereby shifting the balance toward oxidative stress.
3. Section 2.1: This section is over simplified and must be enhanced with text t include descriptions of one-electron (radical) and two-electron (non-radical) oxidants. This is important as the foot print of damage by these different species varies as does the chemical reactivity toward oxidisable biological targets. This section should be broken into subgroups of oxidant species, how they are generated and what type of damage is characteristic of the oxidant being described. Also ONOO- and NO are technically reactive nitrogen species.....
4. IN reference to the list of sources of ROS - page 2, lines 76-77z: This list is not complete and does not include granulocytes of inflamamotry cells nor oxidant generation in cytosolic and extracellualr compartments where oxidants can be generated and accumulate (an example is the bactericidal action of HOCl).
5. In terms of cell signalling capacity this should be tightly linked to the concentration of the ROS and the target cells - for H2O2 this is generally considered to be at the micromolar level and elicit Ca2+ signaling for example; whereas, pathological levels are up to 100-fold higher.
6. Nitrate and nitrite can also be substrates for enzymes that can generate RONs; this should be discussed.
7. Given the topic it is surprising that this section on this particular salivary peroxidase is not expanded more and certainly is an opportunity to make this review more focused to this oral peroxiodase.
8. Oral infection has also been linked to changes in vascular function and vascular disease and this provides a link to systemic disease. Whether it is the systemic inflammation elicited that plays a role if not clear.
Author Response
Thank you very much for your time. We appreciate your valuable comments, the editing of the text was done thoroughly, and we hope we meet your expectation.
1.Some of the text herein is superficial and there needs to be much more balanced description that includes more specific details. The field of redox biology is littered with reviews that fail to provide sufficicent detail on specific ROS, how they are generated and what characteristic damage they can cause in biological systems leaving researchers to accumulate all oxidants under a single banner ROS to simplify things. However, redox regulation is linked to the species being generated and there is specific pathways being elicited and specific damage being caused by different reactive species.
Thank you for the critical comment, we did extensive corrections in the manuscript focusing on extending and deepening provided information. We re-worked parts introduction, section 2.1., 2.2 and added new section 2.4 – Reactive carbonyl species. We also prepared the figure summarizing the information written in the text part.
- Reference to oxidative or reducing stress: This terminology is not used commonly - it is more considered to write increasing oxidative stress and/or limiting endogenous antioxidant capacity thereby shifting the balance toward oxidative stress.
Edited according to the comment.
3.Section 2.1: This section is over simplified and must be enhanced with text t include descriptions of one-electron (radical) and two-electron (non-radical) oxidants. This is important as the foot print of damage by these different species varies as does the chemical reactivity toward oxidisable biological targets. This section should be broken into subgroups of oxidant species, how they are generated and what type of damage is characteristic of the oxidant being described. Also ONOO- and NO are technically reactive nitrogen species.....
Thank your for valuable comment, we significantly extended section 2.1, added new subsections 2.1.1. ROS generation in the human body and 2.1.2 ROS effect on the human body.
- IN reference to the list of sources of ROS - page 2, lines 76-77z: This list is not complete and does not include granulocytes of inflamamotry cells nor oxidant generation in cytosolic and extracellualr compartments where oxidants can be generated and accumulate (an example is the bactericidal action of HOCl).
Done, changes incorporated.
- In terms of cell signalling capacity this should be tightly linked to the concentration of the ROS and the target cells - for H2O2 this is generally considered to be at the micromolar level and elicit Ca2+ signaling for example; whereas, pathological levels are up to 100-fold higher.
Information incorporated
- Nitrate and nitrite can also be substrates for enzymes that can generate RONs; this should be discussed.
We did, please, see the re-worked section 2.2
- Given the topic it is surprising that this section on this particular salivary peroxidase is not expanded more and certainly is an opportunity to make this review more focused to this oral peroxiodase.
Done, please, see the re-worked section 4.4
- Oral infection has also been linked to changes in vascular function and vascular disease and this provides a link to systemic disease. Whether it is the systemic inflammation elicited that plays a role if not clear.
We extended section 10. Cardiovascular diseases (CVDs) highlighting this point
Reviewer 2 Report
Dear authors,
While the subject of the article is interesting, in my opinion the manuscript is not suitable for publication in the International Journal of Molecular Sciences. The information presented in the review is basic, almost common knowledge by now. From the 58 references cited in the article, only 23 are from the last 5 years. Taking into consideration how much progress has been made recently in the field and that the subject has been intensely discussed over the past decades, I do not think that it brings anything new or that it offers a different perspective. Furthermore, there is no presentation of the molecular mechanisms involving the redox status and the included pathologies number. Even the low number of references used in the manuscript shows that the literature was not exhaustively researched. The article is not badly written, although there are several spelling and language errors, but is not enough for a prestigious journal as IJMS with a IF over 6.
Author Response
Dear reviewer,
We took your comments seriously and completely revised the manuscript to make it more in-depth and targeted. We have extended many sections - Introduction, section 2.1. (added 2 subsections), extended section 2.2, created new section 2.4. Reactive carbonyl species, expanded the table, created a figure and added more information to the disease sections. The number of references has increased to 110, of which 59 were from 2017-2022. We hope that after editing, the manuscript can be considered significantly improved.
The language has been re-checked, and corrections have been made.
Round 2
Reviewer 1 Report
The authors have made a major effort to address the primary issues that were raised on the original assessment of their submission. The revised text now makes a greater impact in the field and adds to the currency of the overview presented here.
I have no further issues to raise.
Reviewer 2 Report
Dear authors,
After reading the revised manuscript, I agree that you have improved its quality and will recommend its publication.